# Screening for Hypertension in adolescents living with HIV: Protocol for a cluster randomized trial to improve guideline adherence

**Raphael Adu-Gyamfi** [1]ᴼ*, **Juliana Enos**[2‡], **Kwame Yeboah**[3‡], **Veronika Shabanova**[4], **Nicola Hawley**[4], **Deda Alangea Ogum**[1], **Adwoa Agyei Nkansah**[5], **Elijah Paintsil**[4], **Kwasi Torpey**[1]ᴼ

1 School of Public Health, University of Ghana, Accra, Ghana, 2 Noguchi Memorial Institute for Medical Research, University of Ghana, Accra, Ghana, 3 Department of Physiology, University of Ghana Medical School, Accra, Ghana, 4 School of Public Health, Yale University, New Haven, CT, United States of America, 5 Department of Internal Medicine, University of Ghana Medical School, Accra, Ghana

ᴼ These authors contributed equally to this work.
‡ JE and KY also contributed equally to this work.
* radu-gyamfi2@st.ug.edu.gh

**Data Availability Statement:** No datasets were generated or analysed during the current study. All

## Abstract

### Background

Although AIDS-related deaths have reduced with increased access to antiretroviral care, cardiovascular disease-related morbidities among persons living with HIV are rising. Contributing to this is the higher incidence of Hypertension among Persons Living with HIV. The duration of exposure to the virus and antiretroviral drugs plays a vital role in the pathogenesis, putting perinatally infected children and adolescents at higher risk than behaviorally-infected ones, supporting the calls for increased surveillance of Hypertension among them. Despite the availability of guidelines to support this surveillance, the blood pressure (BP) of adolescents living with HIV (ADLHIV) is not checked during clinical visits. This study aims to assess the effect of a theory-based intervention on healthcare workers' adherence to the guidelines for hypertension screening among adolescents.

### Methods

A multi-facility cluster-randomized study will be conducted. The clusters will be 20 antiretroviral therapy sites in the Greater Accra Region of Ghana with the highest adolescent caseload. Data will be extracted from the folders of adolescents (10–17 years) who received care in these facilities six months before the study. The ART staff of intervention facilities will receive a multicomponent theory of planned behaviour-based intervention. This will include orientation on hypertension risk among ADLHIV, provision of job aids and pediatric sphygmomanometers. Six months after the intervention, the outcome measure will be the change from baseline in the proportion of ADLHIV whose BP was checked during clinical visits. The calculated sample size is 400 folders.

relevant data from this study will be made available upon study completion.

**Funding:** The study has received financial grant support the University of Ghana and the Yale University Academic Partnership for HIV Comorbidity Research Training in Ghana and funded by Fogarty International Center Grant(D43) awarded to EP. The funders have had no role in the design, implementation, analysis or write up of the study.

**Competing interests:** The authors have declared that no competing interests exist.

## Implications of findings

This study will generate evidence on the effectiveness of a multicomponent theory-based intervention for improving the implementation of clinical practice guidelines.

## Trial registration

PACTR202205641023383.

## Introduction

Although AIDS-related deaths have declined with increased access to antiretroviral care, cardiovascular disease-related morbidities among Persons Living with HIV(PLHIV) are rising [1]. Persons living with HIV, compared with uninfected persons, experience a higher incidence of myocardial infarction, heart failure, stroke, and atrial fibrillation [1]. Contributing to this is the higher incidence of hypertension among PLHIV [2–4]. A meta-analysis has demonstrated that 35% of all HIV-infected adults on antiretroviral therapy have hypertension, compared with an estimated 30% of uninfected adults [5]. With a reported incidence of 10–16 (per 1000) per month after ART initiation, the duration of HIV infection is also a significant determinant in the development of hypertension, causing perinatally- and behaviorally- infected adolescents to have higher odds of being hypertensive [6, 7]. Consequently, adolescents living with HIV have a higher incidence of hypertension than their uninfected age and sex-matched peers [8, 9]. Therefore, they are classified as being at moderate risk of hypertension, with a call for increased surveillance [10].

Clinical practice guidelines have been developed to support hypertension surveillance among children and adolescents [11, 12]. Despite the availability of these guidelines, implementation compliance by healthcare workers has generally been poor [13–15]. There are reports of unchecked blood pressure during two-thirds of adolescent clinic visits [16] and about 75% of hypertensive adolescents not being appropriately classified and diagnosed [17]. To understand guideline adherence behaviour among healthcare workers and address gaps for improvement, Davis and Taylor propose the need to be guided by behaviour change models [18]. One model which has helped explore healthcare workers' intentions to adhere to various guidelines is the Theory of Planned Behavior (TPB) [19].

The theory postulates that the likelihood of individuals engaging in a behaviour (e.g., checking the BPs of adolescents) correlates with the strength of their intention to engage in the behaviour. This behavioural intention also represents the person's commitment to act and is itself the outcome of the

i.  person's attitudes toward the behaviour (attitude towards adolescent BP screening),

ii.  person's perception of subjective group norms concerning the behaviour (subjective norms influencing adolescent BP screening), and

iii.  extent to which the person perceives him- or herself to have control concerning the behaviour (perceived behavioural control to check BPs of adolescents) [20].

Available evidence indicates that multicomponent theory-based guideline implementation strategies can improve adherence to recommendations [21–23]. However, there is little data in developing countries, including Ghana, on the implementation of these strategies and their effect on improvement on healthcare worker adherence to hypertension screening and

management guidelines for adolescents.Thus, the study aims to assess the effect of a multicomponent theory of planned behaviour-based intervention on the adherence of antiretroviral therapy clinics to the guidelines for screening and managing Hypertension among adolescents living with HIV.

## Methods

### Study settings and participants

The study will be conducted according to the Standard Protocol Items: Recommendations for interventional Trials(SPIRIT) Guidance which is attached as a supporting information (S1 Checklist). It will be conducted in selected antiretroviral therapy facilities in the Public Health sector in the Greater Accra Region, where 28% of the adolescents living with HIV in Ghana receive care using a parallel group cluster randomized design. Currently there are 58 facilities providing antiretroviral therapy services in the Region. Of these 58 facilities, those with a minimum of 40 ADLHIV on ART, as evidenced from their clinic database as of December 2021 will be included in the study.

The study population will include all health care workers(HCW) who provide clinical services for PLHIV at the facilities that will be selected for the study.

However, facilities that do not consent to participate and those participating in similar studies involving the ART staff will not be included.

### Participant selection

Recruitment into the study will be done at two levels: health facility and health workers.

**a. Health facility recruitment.** Data from DHIS will be usedto identify HIV service provision sites that have PLHIV aged 10–19 years in care. Those with more than 40 adolescents in treatment will be eligible for inclusion in the study. Management of prospective facilities will be formally engaged about the study and permission sought for facility inclusion. Consenting facilities will be requested to provide a focal person who will coordinate the study activities within the facility.

**b. Clinical staff recruitment.** Once facilities are recruited, the PI will contact the focal persons to obtain details of potential clinical staff interested in participation. All HCW aged 18 years and above, and have valid licenses from relevant regulatory bodies in Ghana will be eligible for inclusion in the study. They will include prescribers, Pharmacy staff, Nurses and Midwives. They will, however, be excluded if

- HCW has less than one year experience providing clinical care services to PLHIV

- They are not available throughout the study period.

### Sample size

This study is designed to detect a doubling (100%) of the proportion of adolescent visits that had BP screening done (conservatively calculated in STATA version 16 as an increase from 17% to 34%) between the intervention and control groups. It is based on the assumption of a 2-sided test with 80% statistical power and an alpha level of 0.05, cluster sizes of 20 adolescent records per facility and an intra-cluster correlation coefficient (ICC) of 0.05.

From the STATA output, 20 health facilities (10 each in the intervention and control arms) will be selected. Twenty adolescent records will be assessed per health facility, giving a total of 400 adolescent records from all the 20 health facilities. With a reported average of 4 healthcare

workers providing antiretroviral care per site in the Greater Accra Region, a total of 80 eligible and consenting healthcare workers in the participating facilities will be involved in the study.

## Randomization

**a. Sequence generation.** The unit of randomization will be the health facility. Randomization will be performed by a statistician independent of the study team, using computer-generated random permuted blocks. It will be stratified by type of health facility (regional, district/municipal, Polyclinic, Health Centre). Ten facilities will be in the intervention arm and the rest in the control arm. The study statistician will provide each facility with a computer-generated list of random numbers to select a sample of 20 adolescent client records to be reviewed at the baseline and follow-up.

**b. Allocation concealment.** Each facility will eventually become aware of its allocation, but allocation will be concealed until the baseline assessment is completed.

**c. Implementation of clusters.** The PI will provide the list of prospective facilities by type to the study statistician anonymized by an ID. The study statistician will generate the random sequence and allocate practices to two separate dummy-coded groups without knowing which group will receive the intervention.

**d. Blinding.** Given the nature of the intervention, healthcare workers will inevitably be aware of their allocation and thus, blinding participants will not be possible. Each facility will also become aware of allocation. However, facilities will not be informed of allocation until after both facilities and individual health workers are identified and recruited (i.e., not until after baseline assessment). Adolescents within each facility will remain blind to allocation. The outcome assessors will be kept blind to the allocation. The facility contact using the random sequence of numbers to identify the sample of 20 patient records within each facility will possibly not be blind to allocation. Blinding the entire research team is impossible, as intervention facilities will be contacted to arrange the sessions. However, the study statistician conducting the outcome analysis will remain blinded to allocation until after the data analysis [24].

## Intervention

The ten intervention facilities will receive a multicomponent intervention package based on the theory of planned behaviour, as captured in Table 1. They will be oriented on the risk of Hypertension among adolescents living with HIV, be provided monthly feedback on their performance for the initial three months and receive the support of an opinion leader in their facility. They will also receive clinical decision support and sphygmomanometers with paediatric cuffs. The Fidelity of intervention delivery will be assessed using the process measures captured in Table 1 across the TPB domains.

## Comparator

All 20 facilities will be given an orientation on the guidelines for screening and managing Hypertension among children and adolescents. Training facilitators will use PowerPoint presentations, group discussions and case scenarios to do this.

## Follow up

Both intervention and control facilities will be followed up for six months after the orientation. The flow of the study procedures is as captured in Fig 1.

**Table 1. Application of theory of planned behaviour to intervention design.**

| TPB Constructs | Intervention package | Process Measures |
|---|---|---|
| Attitude | 1. Orientation on hypertension risk among adolescents living with HIV | Pre and post-test scores |
| 1. Subjective norms | 1. Audit and feedback<br>2. Opinion leaders<br>3. Patient education<br>4. Reminders(posters) | 1. The proportion of planned feedback sessions undertaken<br>2. The proportion of planned engagement sessions with opinion leaders held<br>3. The proportion of planned patient education sessions held<br>4. The proportion of expected posters pasted |
| Perceived behavioural control | 1. Provision of a paediatric sphygmomanometer<br>2. Orientation on BP measurement technique<br>3. Provision of clinical decision support | 1. The proportion of intervention facilities with a paediatric sphygmomanometer<br>2. a. proportion of target staff oriented on BP measurement technique<br>3. b. change in accuracy of BP measurements post-orientation<br>4. The proportion of intervention facilities with clinical decision support pack |

## Outcomes

The effect of the intervention on healthcare workers' adherence to the guideline will be assessed using

i. frequency of screening (primary)

ii. Diagnosis of elevated or high BP (secondary),

iii. investigation of high BP (secondary) and

iv. management of elevated or high BP (secondary).

## Data collection

Data will be collected using a data extraction tool based on Ghana's HIV client care booklet. The tool will extract the data from the client care records of adolescents who received care within the period of interest. At baseline, it will assess the records of 20 randomly selected adolescents who received care at the facility six months before the assessment and extract the data from their most recent visit. At follow-up, data will be extracted from the most recent visit of 20 randomly selected adolescents who received care at the facility after the intervention commenced.

## Data management

**a. Data quality control and access.** Qualified research assistants will be trained in collecting data by conducting pre-tests of the data extraction collection tools. The PI will cross-check all data entry forms to ensure errors are corrected. All data will be entered in duplicate to reduce data entry errors. Hard copies of data will be stored under lock and key. Soft copies of data will be stored in a One Drive account that will be accessible to the principal investigator only after completion of the study.

The data will be analyzed and published as conference abstracts, posters and presentations. Manuscripts will also be developed for publication from the study findings. All of the individual participant data collected during the study, after deidentification, will be available for sharing. The study protocol, statistical analysis plan, informed consent form, and analytic codes will also be available. The data will be available beginning two months and ending ten years following the study to anyone who wishes to access the data for individual participant meta-analysis. Requests for the data should be directed at radu-gyamfi2@st.ug.edu.gh.

| ACTIVITY | 1 | 2 | 3 | 4 | 5 | 6 | 7 | 8 | 9 | 10 |
|---|---|---|---|---|---|---|---|---|---|---|
| **TIMEPOINT (Months)** | Enrolment and allocation | | | Intervention | | | | | | End line |
| **ENROLLMENT:** | ▩ | ▩ | | | | | | | | |
| Eligibility screen | ▩ | ▩ | | | | | | | | |
| Informed consent | ▩ | | | | | | | | | |
| Allocation | | | ▩ | | | | | | | |
| **INTERVENTIONS:** | | | | ▩ | ▩ | ▩ | ▩ | ▩ | ▩ | |
| Orientation on hypertension risk among adolescents living with HIV | | | | ▩ | | | | | | |
| Audit and feedback | | | | ▩ | ▩ | ▩ | | | | |
| Opinion leaders support | | | | ▩ | ▩ | | ▩ | ▩ | ▩ | |
| Patient education | | | | ▩ | ▩ | ▩ | ▩ | ▩ | ▩ | |
| Reminders(posters) | | | | ▩ | ▩ | ▩ | ▩ | ▩ | ▩ | |
| Provision of a paediatric sphygmomanometer | | | | ▩ | | | | | | |
| Orientation on BP measurement technique | | | | ▩ | | | | | | |
| Provision of clinical decision support | | | | ▩ | | | | | | |
| **ASSESSMENTS:** | | | | | | | | | | |
| Frequency of screening (primary) | ▩ | ▩ | | | | | | | | ▩ |
| Diagnosis of elevated or high BP | ▩ | | | | | | | | | ▩ |
| Investigation of high BP (secondary) | ▩ | | | | | | | | | ▩ |
| Management of elevated or high BP | ▩ | ▩ | | | | | | | | ▩ |

**Fig 1. Schedule of enrolment, interventions and assessments.**

**b. Data analysis.** Characteristics of healthcare facilities, providers and patients will be summarized using descriptive statistics, such as means (standard deviation) and medians (25th percentile, 75th percentile) for continuous variables and frequency (percent) for categorical variables. Visual examinations of histograms will be used to check for skewness and identify potential outliers among continuous variables. The primary binary outcome of the prevalence of BP screening will be analyzed using a two-level hierarchical Poisson regression modelling, where level 1 is the patient and level 2 is the health facility (level 2 random intercept). There will be two fixed effects: time (baseline, follow-up) and intervention group (yes/no) and their interaction. The time effect will examine whether the prevalence of BP screening changes across time regardless of intervention. The intervention effect will examine whether, regardless of time, the intervention group differed from the control group. The interaction effect will provide the between-group difference of the within-group change in BP screening prevalence from baseline to follow-up. Results will be summarized using estimated proportions (prevalence) of BP screening with surrounding 95% Confidence Intervals (95%CIs), as well as Prevalance Ratios (PR) and surrounding 95%CIs. Secondary outcomes, which are also rates, will be analyzed similarly. All tests will be two-tailed and conducted at the alpha level of 0.05. Statistical analyses will be implemented using STATA 16.

## Ethical considerations

Ethical approval has been received from the Ghana Health Service Ethics Review Committee (GHS-ERC: 013/05/22). Permission will be sought from the Regional and respective Municipal Health Directorates of the study areas and the Managers of the selected Health Facilities. Permission will also be obtained from the heads of Medical Records unit in the respective facilities before extracting data from the client care booklets.

All prospective study participants will be made aware that participation in the study is entirely voluntary and that they are free to drop out of the study (and can choose not to disclose reasons for dropping out) at any time with no negative consequence. All participants will sign a written informed consent before enrolment into the study. The consent form will describe the aim of the study, procedures, risks, benefits and compensation. The investigator will be available to answer all questions and allow time for potential participants to think through them and make a decision.

The study has been registered with the Pan African Clinical Trial Registry with Registration Number **PACTR202205641023383**. The detailed protocol is attached as a supporting information(S1 Protocol).

## Discussion

The study will be a two-arm cluster randomized trial assessing the comparative effectiveness of the usual implementation of the guideline for screening and managing hypertension among adolescents against using a theory-based, multicomponent strategy to implement the same guideline. The intervention combines conventional orientation on the clinical practice guidelines with a theory of planned behaviour-based intervention package. The primary analysis will compare the change from the baseline of the proportion of adolescents with BP checked between the intervention and control facilities.

The study's findings are expected to provide evidence to guide health system strengthening efforts towards preventing cardiovascular diseases, particularly Hypertension. It is expected to generate scientifically valid information to guide frontline health workers, health facility managers, health policymakers, pre-service and in-service curriculum developers for relevant cadre of health workers. With the reported paucity of similar studies [25, 26], it will be one of the

few, if not the first study from Ghana, to contribute to the literature on the effect of theory-based strategies targeting the implementation of clinical practice guidelines.

The adherence of the healthcare workers in intervention facilities to the intervention protocol is an anticipated challenge to the intervention delivery. Efforts will, however, be made to explain the study procedures to all prospective participants at enrollment and during follow-up engagements.

## Supporting information

**S1 Checklist. SPIRIT checklist.**
(PDF)

**S1 Protocol. Research protocol.**
(DOCX)

## Author Contributions

**Conceptualization:** Raphael Adu-Gyamfi, Juliana Enos, Kwame Yeboah, Deda Alangea Ogum, Elijah Paintsil, Kwasi Torpey.

**Methodology:** Raphael Adu-Gyamfi, Juliana Enos, Kwame Yeboah, Veronika Shabanova, Nicola Hawley, Elijah Paintsil, Kwasi Torpey.

**Supervision:** Juliana Enos, Kwame Yeboah, Kwasi Torpey.

**Writing – original draft:** Raphael Adu-Gyamfi.

**Writing – review & editing:** Raphael Adu-Gyamfi, Kwame Yeboah, Veronika Shabanova, Adwoa Agyei Nkansah, Elijah Paintsil, Kwasi Torpey.

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
