## [Decision Letter · Decision Letter 0]

26 Jun 2023

PONE-D-23-13757Screening for Hypertension in adolescents living with HIV: protocol for a cluster randomized trial to improve guideline adherencePLOS ONE

Dear Dr. Adu-Gyamfi,

Thank you for submitting your manuscript to PLOS ONE. After careful consideration, we feel that it has merit but does not fully meet PLOS ONE’s publication criteria as it currently stands. Therefore, we invite you to submit a revised version of the manuscript that addresses the points raised during the review process.

ACADEMIC EDITOR: After a careful evaluation of your manuscript and the review comments, you are required to address the issues raised by the reviewers and re-submit your revised manuscript. You can disregard reviewer # 1 comment on lines 31, 86 and 87 asking if the study is already done or yet to be done.

We look forward to receiving your revised manuscript.

Kind regards,

Stephen Apanga, PhD

Academic Editor

PLOS ONE

3. We note that the original protocol that you have uploaded as a Supporting Information file contains an institutional logo. As this logo is likely copyrighted, we ask that you please remove it from this file and upload an updated version upon resubmission.

Additional Editor Comments:

You are invited to address the comments raised by the reviewers and re-submit your revised manuscript.

Reviewers' comments:

Reviewer's Responses to Questions

**Comments to the Author**

1. Does the manuscript provide a valid rationale for the proposed study, with clearly identified and justified research questions?

Reviewer #1: Yes

Reviewer #2: Yes

2. Is the protocol technically sound and planned in a manner that will lead to a meaningful outcome and allow testing the stated hypotheses?

Reviewer #1: Yes

Reviewer #2: Yes

3. Is the methodology feasible and described in sufficient detail to allow the work to be replicable?

Reviewer #1: Yes

Reviewer #2: Yes

4. Have the authors described where all data underlying the findings will be made available when the study is complete?

Reviewer #1: Yes

Reviewer #2: Yes

5. Is the manuscript presented in an intelligible fashion and written in standard English?

Reviewer #1: Yes

Reviewer #2: Yes

6. Review Comments to the Author

You may also provide optional suggestions and comments to authors that they might find helpful in planning their study.

Reviewer #1: Review Comments:

Abstract

Line 52; ‘them’ refers to what exactly

Introduction

Iine 78, seems to indicate available evidences to your study, why doing it.

Methods

Line 31,86 and 87 ; Is the study already done or yet to be done. In either case check your language in the entire manuscript.

Inclusion criteria must be properly be stated to capture age group above 18 years. What happens to facilities that provide the antiretroviral drugs services? What kind of qualification is accepted?

Line 104&105, Why not added to inclusion criteria section.

What is the difference between setting & recruitment and inclusion & exclusion criteria since the contents are similar.

Line 110 to 112, 114 and 115; the language seems more of proposal rather than article for publication.

Line 124; how did you come by 10, 80 and 400?

What type of blinding was done?

Line 166; Yet to be followed or had been followed?

Line 180 & 189, was the data extracted from client records or questionnaire was used? Needs clarification.

What becomes of the data after the study?

What is new about this study

Reviewer #2: This is the description of the protocol of an important multi-facility cluster-randomized study which will be conducted in 20 antiretroviral therapy sites in the Greater Accra Region of Ghana in which adolescent living with HIV are routinely seen for care. The protocol is very well written and includes detailed descriptions of randomisation, concealment of allocation, blinding and statistical procedures as requested by the CONSORT statement. The study if completed is likely to provide evidence to guide health system strengthening efforts towards targeting missing opportunities for monitoring of hypertension and preventing cardiovascular disease in adolescents living with HIV. I support the publication of trial protocols as a tool to increase reproducibility in medical research.

I only have minor comments:

Line 103. 28% of the adolescents living with HIV in Ghana receive care in this Region does not seem to be ‘Most’ of them.

Lines 178-185. I assume this will be done at each 20 facilities to achieve the target sample size of 20 x 20 = 400 adolescents. It is unclear from reading the text.

Line 192. The fact that the principal investigator will have access to the data is an anomaly. I suggest that especially the PI is kept blind to the data as the knowledge of interim data could affect the conduct of the trial.

Line 198. I would avoid square root transformations as it is cumbersome to transform back to the original scale. Use non-parametric tests on the natural scale instead.

Line 208. If a logistic regression is used it should be Odds Ratios or Risk Ratios, not Rate Ratios

7. PLOS authors have the option to publish the peer review history of their article (what does this mean?). If published, this will include your full peer review and any attached files.

Reviewer #1: No

Reviewer #2: No

---

## [Author Response · Author response to Decision Letter 0]

16 Aug 2023

Reviewer 1 Comments and responses

Abstract

Comment: Line 52; ‘them’ refers to what exactly.

Response: Please, the “them” refers to Persons Living with HIV(PLHIV) The sentence has been made clearer and now reads as follows: “ Contributing to this is the higher incidence of hypertension among Persons Living with HIV”.

Introduction

Comment: Iine 78, seems to indicate available evidence to your study, why doing it.

Response: The reason for conducting this study has kindly been added to the introduction section as “Available evidence indicates that multicomponent theory-based guideline implementation strategies can improve adherence to recommendations[21][22,23]. However, there is little data in developing countries, including Ghana, on the implementation of these strategies and their effect on improving healthcare worker adherence to hypertension screening and management guidelines for adolescents. Thus, the study aims to assess the effect of a multicomponent theory of planned behaviour-based intervention on the adherence of antiretroviral therapy clinics to the guidelines for screening and managing hypertension among adolescents living with HIV.”

Methods

Comment 1 : Inclusion criteria must be properly be stated to capture age group above 18 years. What happens to facilities that provide the antiretroviral drugs services? What kind of qualification is accepted?

Response: The authors consider these recommendations valuable and the session for study setting, population and participant selection have been revised to capture them. 

Comment 2: Line 104&105, Why not added to inclusion criteria section.

What is the difference between setting & recruitment and inclusion & exclusion criteria since the contents are similar.

Response: Thanks for noting this. The authors have revised the whole section and added them to the inclusion and exclusion criteria

Comment: Line 110 to 112, 114 and 115; the language seems more of proposal rather than article for publication.

Response: Please the manuscript is a study protocol, which is yet to be conducted

Comment: Line 124; how did you come by 10, 80 and 400?

Response: Please the section under sample size has been reworded to make this clearer. 

Comment: What type of blinding was done?

Response: The study will be double blinded. This is because, although the participants will be aware of their allocation, the outcome assessors (data collectors) and the study statistician (data analyst) will be kept blind to the allocation of the participants, as described in that section.

Comment: Line 166; Yet to be followed or had been followed?

Response: Please the study is yet to be conducted so they are yet to be followed.

Comment: Line 180 & 189, was the data extracted from client records or questionnaire was used? Needs clarification.

Response: Thank you for noting this. A data extraction tool will be used to obtain data from client records. The correction has been made and now reads:

“Qualified research assistants will be trained in collecting data by conducting pre-tests of the data extraction tools.”

Comment: What becomes of the data after the study?

Response: Please the Data Quality Control and Access section of the manuscript has been revised to include this information.

Comment: What is new about this study

Response: As at the time of manuscript development, this is the first protocol of a study to assess the effect of theory-based interventions on guideline adherence among healthcare workers in Ghana. This has been stated in the discussion section of the manuscript. 

Reviewer 2 Comments and Responses

Comment: Line 103. 28% of the adolescents living with HIV in Ghana receive care in this Region does not seem to be ‘Most’ of them.

Response: Thanks for the comment. Please this sentence has been revised and now reads as follows:

“ The study will be conducted in selected antiretroviral therapy facilities in the Public Health sector in the Greater Accra Region, where 28% of the adolescents living with HIV in Ghana receive care”.

Comment: Lines 178-185. I assume this will be done at each 20 facilities to achieve the target sample size of 20 x 20 = 400 adolescents. It is unclear from reading the text.

Response: Please this comment has been considered and the section under Sample Size has been revised to make this clearer.

Comment: Line 192. The fact that the principal investigator will have access to the data is an anomaly. I suggest that especially the PI is kept blind to the data as the knowledge of interim data could affect the conduct of the trial.

Response: The authors consider this comment very relevant. The PI’s supervisors will have access to the electronic version of the data and access will only be granted to the PI after the study is completed. The section under Data Quality Control and Access has been revised to reflect this. 

Comment: Line 198. I would avoid square root transformations as it is cumbersome to transform back to the original scale. Use non-parametric tests on the natural scale instead.

Response: The study team has considered the recommendation and agreed that there won't be any data transformation. That statement has therefore been deleted from the data analysis section.

Comment: Line 208. If a logistic regression is used it should be Odds Ratios or Risk Ratios, not Rate Ratios. 

Response: The comment has been considered by the study team. Poisson regression will rather be conducted instead of logistic regression. Prevalence ratios, instead of rate ratios will also be determined. The section under data analysis has thus been revised to reflect this change.

---

## [Decision Letter · Decision Letter 1]

27 Mar 2024

Screening for Hypertension in adolescents living with HIV: protocol for a cluster randomized trial to improve guideline adherence

PONE-D-23-13757R1

Dear Dr. Adu-Gyamfi,

We’re pleased to inform you that your manuscript has been judged scientifically suitable for publication and will be formally accepted for publication once it meets all outstanding technical requirements.

Kind regards,

Miquel Vall-llosera Camps

Staff Editor

PLOS ONE

Reviewers' comments:

Reviewer's Responses to Questions

**Comments to the Author**

1. Does the manuscript provide a valid rationale for the proposed study, with clearly identified and justified research questions?

Reviewer #3: Yes

2. Is the protocol technically sound and planned in a manner that will lead to a meaningful outcome and allow testing the stated hypotheses?

Reviewer #3: Yes

3. Is the methodology feasible and described in sufficient detail to allow the work to be replicable?

Reviewer #3: Yes

4. Have the authors described where all data underlying the findings will be made available when the study is complete?

Reviewer #3: Yes

5. Is the manuscript presented in an intelligible fashion and written in standard English?

Reviewer #3: Yes

6. Review Comments to the Author

You may also provide optional suggestions and comments to authors that they might find helpful in planning their study.

Reviewer #3: The authors adequately addressed all the previous comments raised by the reviewers. I have no more comments to add.

7. PLOS authors have the option to publish the peer review history of their article (what does this mean?). If published, this will include your full peer review and any attached files.

Reviewer #3: No
